# Developing and validating a dynamic model of water production by direct-contact membrane distillation

Emad Ali[1]*, Jamel Orfi[2], Abdullah Najib[2]

**1** Department of Chemical Engineering, King Saud University, Riyadh, Saudi Arabia, **2** Department of Mechanical Engineering, King Saud University, Riyadh, Saudi Arabia

These authors contributed equally to this work.
* amkamal@ksu.edu.sa

**Data Availability Statement:** All relevant data are within the manuscript and Supporting Information files.

**Funding:** The research is funded by the Deanship of Scientific Research, King Saud University

## Abstract

We consider the development and fitting of a dynamic model for desalinated water production by a direct-contact membrane distillation (DCMD) unit. Two types of dynamic-model structures, namely, lumped parameter and spatial, were evaluated. Both the models were validated using experimental response data generated by step testing the inlet hot stream temperature of a DCMD pilot plant. Both the model structures failed to follow the dynamic response adequately. However, a modification of the model by adding a heat loss term resulted in enhanced predictions for both model structures. The overall relative error in the model–plant mismatch was approximately 3%. This is reasonable considering the random uncertainties associated with the plant operation. This observation also improves our understanding of the importance of using better correlations for heat-transfer coefficients, to develop a more reliable and accurate predictive model for a wide range of operating conditions.

## Introduction

The shortage of potable water is a major problem persisting in several regions worldwide. The growth and development of the desalination industry are remarkable. In 2016, global desalination production attained approximately $85 \times 10^6$ m³/day [1]. The conventional desalination technologies, namely, multi-stage flash, multi-effect evaporation, and reverse osmosis, are known for being energy intensive. Among the most innovative and potential desalination technologies, membrane distillation (MD) is gaining interest because of its advantages compared to conventional technologies. The direct-contact membrane distillation (DCMD) configuration, which is a widely employed MD configuration, is known for its attractive characteristics such as its requirement of low operating temperature and hydrostatic pressure. It can achieve approximately 100% rejection of salt ions. Moreover, its permeate quality is marginally affected by the feed concentration. Furthermore, it has a compact and flexible structure [2–5]. Notwithstanding the appealing features of the MD technology, its industrial commercialization is hindered by certain technical barriers and deficiencies, such as low recovery ratio; high specific

through Research Group no (RG- VPP 091). The funder had no role in study design, data collection and analysis, decision to publish, or preparation of the manuscript.

**Competing interests:** The authors have declared that no competing interests exist.

energy requirements; and fouling and scaling, which eventually deteriorate the permeate flux and membrane pore wetting [2,6–8]. Therefore, several investigations were performed during the past years to enhance the feasibility of the MD technology as a commercial desalination technology [9–11]. For example, many researchers focused on improving the MD performance by performing reconfiguration or retrofitting such as incorporation of heat-recovery systems [10,12], recycling of the discharged brine to the inlet stream [13,14], and use of multi-staging [10,15]. Integration of the MD technology with low-grade energy sources such as waste heat, solar, and geothermal energy has been addressed [16–18]. To address science and engineering issues that limit the development and commercialization of the MD technology, theoretical investigations concerning the design and optimization of the MD process have been performed [3,19–26]. These theoretical investigations, which modeled several physical principles integral to the MD process, are subject to the following limiting assumptions:

- Unlike the studies on steady-state regime, those on transient conditions are very few [21,27].

- Heat and mass transfers are generally space independent (using lumped approach) or one dimensional [5,20]. Only a few works have been concerned with refined multidimensional aspects [28–30].

- The transfer coefficients are generally evaluated using inappropriately developed correlations, e.g., for impermeable interfaces and not for permeable ones [31,32]. A deeper perception of the heat- and mass-transfer phenomena should be addressed.

- It has been reported that the uncertainty of membrane properties causes inadequate mass fluxes. Camacho et al. [33] reported that membrane properties such as porosity, thickness, and pore size may change because of the membrane compression caused by the hydrodynamic pressure associated with the circulating feed and permeate flow rates. As mentioned by Andrjesdóttir et al. [9], the simultaneous fitting of heat and mass data to a first-principles model can aid the understanding of the underlying physics of the MD process and, thereby, highlight the deficiencies of similar physical models.

Irrespective of the effectiveness and complexity of these models, they are limited and cannot be utilized for applications such as automation and control implementation because they are stationary. In addition, these models cannot be integrated with time-varying energy resources such as solar and wind energies, which are characterized by intermittent energy outputs. Therefore, it is important to develop dynamic models that consider the fluctuation or abrupt variations in energy supply. Charfi et al. [28] undertook one of the earliest efforts to address the dynamics modeling of the MD processes. Hassan et al. [34] developed a dynamic, spatial finite-difference model to investigate the transient performance of a vacuum MD when hot feed parameters such as mass-flow rate, temperature, and concentration distributions undergo variations in steps.

Recently, Eliewi et al. and Karam et al. [21,27] proposed DCMD-based dynamic models that consider heat and mass variations both in one- and two-dimensional spaces. These space-dependent models are suitable for exploring the impact of hidden unmeasured parameters. However, for control applications and/or integrating the MD technology with fluctuating energy sources, the dynamics of the external inputs and outputs are important. Moreover, in the works of Eliewi et al. and Karam et al. [21,27], the dynamics of a single-process output, i.e., the permeate outlet temperature, is validated, whereas those for the brine outlet temperature is omitted. Furthermore, the validation is based on the ramp variations in the temperature. Furthermore, this type of dynamics is not common for the MD process, and common dynamic capturing should be based on step testing because it defines both the transient and stationary

behaviors of the MD process. Recently, Ali [35] developed a dynamic model for the outlet cold and hot temperatures in a DCMD. The model is converted to a transfer function (TF) to analyze the dynamic characteristics of the process outputs.

This study is a continuation of a previous work [35] by performing dynamic analysis and modeling of the DCMD process. Specifically, the dynamics of mass production is modeled, because it is the main product of the process. To the authors' knowledge, no detailed derivation or validation of dynamic model of mass flux in DCMD has been published. All the published studies address the modeling of only the dynamics of the outlet temperatures of the MD process. Furthermore, a detailed derivation of the explicit formulation of the mass flux dynamics is presented. The generated model is based on first principles that include the internal physics of the MD process. The measured dynamic data of the mass accumulation in the DCMD plant for water desalination are analyzed and correlated to obtain a typical transient behavior. Standard reaction-curve methods [36] are utilized to infer the dynamic characteristics of the mass production response from the correlated transient responses. Moreover, the correlated transient behavior obtained is used to validate the theoretical dynamics model. The analysis here is limited to the DCMD module using a spiral-wound membrane made of polyethylene tetrafluoride. The analysis covers a wide range of operating conditions for the mass-flow rate and inlet hot temperature.

## Experimental setup description

The development and validation of the model are based on the experimental data generated from an MD pilot plant. The pilot plant was developed by SolarSpring [37] and equipped with a DCMD module with an effective membrane-area of 10 m$^2$, membrane thickness of 230 μm, channel length of 14 m, channel height of 0.7 m, pore diameter of 0.2 μm, and channel gap of 2 mm. The membrane porosity is 0.8, and the water-entry pressure is 4.1 bar. A schematic of the MD process is depicted in Fig 1. A data-acquisition system is employed for data logging and for regulating various instruments. An external electrical heater (H1) is used to heat the evaporator circuit, i.e., the membrane's hot feed stream. The hot feed temperature is controlled effectively using a programmable logic controller. An external cooler (H2) is also incorporated to adjust the temperature of the MD inlet cold stream. The temperature of the inlet cold stream is regulated manually. Hence, the inlet cold temperature undergoes fluctuation and disruptions because of the fuzziness of manual control. The desalinated water is separated through an overflow, accumulated in the storage tank T3, and measured using an electronic balance. Further details of the experimental setup and procedure are available in previous works [32,35,38,39]. In the previous works, the experimental device was used to generate data to be used for calibrating a steady state MD model, conducting energy and exergy analysis, and validating a dynamic model of the MD outlet temperatures. In this work, the time-evolution measurement of the mass flux is used to validate the proposed dynamic model for mass production. Note that in all the experiments, the mass-flow rates both on the hot and cold side are maintained equal. This is because the experimental module does not allow for non-equal flow rates on the two sides, to avoid membrane sheet deformation.

## Mass-flux dynamic model

The water distillate is the principal output of the process. The distillate production undergoes transient behavior during the startup or stepped variations in the process inputs such as feed temperature and/or feed-flow rate. Hence, we intend to develop a theoretical model that captures the dynamic behavior of mass production. We assume an absence of heat loss to the environment, as well as stable membrane properties such as thickness, tortuosity, porosity, and

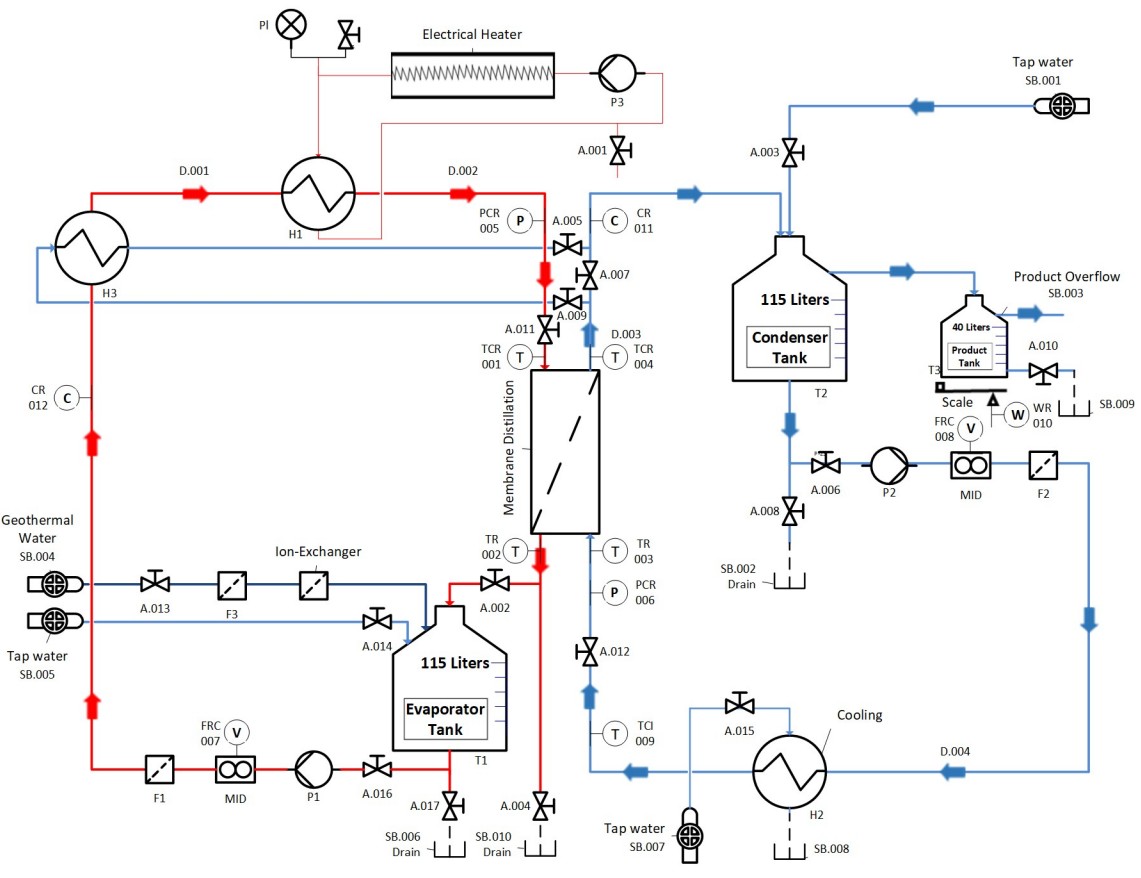

**Fig 1. Module test facility (flowsheet).**

pore size. In addition, we assume the total pressure drop across the membrane to be negligible. Furthermore, the physical properties of water such as density, heat capacity, thermal conductivity, viscosity, and heat-transfer coefficient are functions of temperature. The dynamic model for mass production of water in the lumped-parameter case is expressed by the following initial-value problem (IVP):

$$\frac{dm_w}{dt} = \frac{C_m A}{\left(1 + C_m \beta_h c_h H_v + C_m \beta_c c_c H_v\right)} \left( (\beta_h a_h - \beta_c b_h)\frac{1}{2}\sqrt{\frac{T_{h_{in}}}{T_{h_{out}}}}\frac{dT_{h_{out}}}{dt} + (\beta_h a_c - \beta_c b_c)\frac{1}{2}\sqrt{\frac{T_{c_{in}}}{T_{c_{out}}}}\frac{dT_{c_{out}}}{dt} \right) \quad (1)$$

$$v\rho Cp\frac{dT_{h_{out}}}{dt} = m_{h_{in}}Cp\left(T_{h_{in}} - T_{rf}\right) - m_{h_{out}}Cp\left(T_{h_{out}} - T_{rf}\right) - h_m A\left(T_{h_m} - T_{c_m}\right) - j_w A H_v \quad (2)$$

$$v\rho Cp\frac{dT_{c_{out}}}{dt} = m_{c_{in}}Cp\left(T_{c_{in}} - T_{rf}\right) - m_{c_{out}}Cp\left(T_{c_{out}} - T_{rf}\right) + h_m A\left(T_{h_m} - T_{c_m}\right) + j_w A H_v \quad (3)$$

The derivation of the mass production dynamic (Eq 1) and the definitions of the underlying parameters are provided in Appendix A. The thermal Eqs (2) and (3) are adopted from Karam et al. [27] and presented and discussed in another work [40]. The thermal equations are developed by applying the energy conservation law on the entire MD module (S1 Fig) both on the feed and permeate sides. In due course, the accumulation of energy within the hot channel becomes equal to the difference between the heat loss from the hot fluid as sensible heat and the heat transferred to the cold side. Within the cold side, the accumulation of energy is equal

to the difference between the heat gained by the permeate fluid and the heat transferred from the hot side. The heat transfer between the two sides occurs through two mechanisms: conduction through the membrane material and latent heat of vaporization. The initial conditions for the IVP are that the mass production is zero, and the bulk temperature is equal to the room temperature. The reference temperature ($T_{rf}$) is also considered to be equal to the room temperature. The IVP is solved simultaneously to obtain the time evolution of mass production. Although the above-mentioned dynamic model appears linear in its structure, nonlinearity arises from the membrane heat-transfer coefficient ($h_m$), membrane coefficient ($C_m$), membrane surface temperature ($T_{h_m}$, $T_{c_m}$), and underlying parameters ($a$'s, $b$'s, and $\beta$'s). These coefficients and parameters are implicitly functions of the bulk temperature, membrane interface temperatures, and mass flux. Hence, these variables are calculated using an iterative procedure assuming the pseudo-steady state specified in Appendix B (see Algorithm S1). Notably, the thermal behavior of the process is represented by the boundary temperatures, which are measured and therefore, verifiable. It is apparent from Eqs (1)–(3) that the dynamic of the mass production follows that of the outlet temperatures. Note that an equal inlet feed-flow rate ($m_{h_{in}} = m_{c_{in}}$) is used for the hot and cold sides, which is imposed by the experimental setup.

For long membrane modules, the lumped-parameter model is limited because it uses a constant value for the temperature profile along the membrane length. A better representation of the process behavior can be achieved by using spatial model formulation. In this case, the process variables such as the bulk temperature and mass flux are permitted to vary over the membrane length. Hence, the dynamic model of the MD (Eqs (1)–(3)) can be written for each control element (S1 Fig) (denoted by subscript $i$) along the membrane length, using the lumped-capacitance method as follows:

$$\left(\frac{v}{n}\right)\rho Cp\frac{dT_{h_i}}{dt} = m_{h_{i-1}}Cp\left(T_{h_{i-1}} - T_{rf}\right) - m_{h_i}Cp\left(T_{h_i} - T_{rf}\right) - h_{m_i}\,\Delta xh\left(T_{h_{m,i}} - T_{c_{m,i}}\right) - j_{w_i}\,\Delta xhH_v \quad (4)$$

$$\left(\frac{v}{n}\right)\rho Cp\frac{dT_{c_i}}{dt} = m_{c_{i+1}}\;Cp\left(T_{c_{i+1}} - T_{rf}\right) - m_{c_i}Cp\left(T_{c_i} - T_{rf}\right) + h_{m_i}\,\Delta xh\left(T_{h_{m,i}} - T_{c_{m,i}}\right) + j_{w_i}\,\Delta xhH_v \quad (5)$$

$$\frac{dm_{w_i}}{dt} = \frac{C_{m_i}A}{(1 + C_{m_i}\beta_{h_i}c_{h_i}H_{v_i} + C_{m_i}\beta_{c_i}c_{c_i}H_{v_i})}\left(\left(\beta_{h_i}a_{h_i} - \beta_{c_i}b_{h_i}\right)\frac{dT_{h_i}}{dt} + \left(\beta_{h_i}a_{c_i} - \beta_{c_i}b_{c_i}\right)\frac{dT_{c_i}}{dt}\right) \quad (6)$$

$i = 1, \cdots, n$

$$\text{for } i = 1 \;\rightarrow\; T_{h_{i-1}} \equiv T_{h_{in}} \;;\; T_{c_i} \equiv T_{c_{out}}$$

$$\text{for } i = n \;\rightarrow\; T_{h_i} \equiv T_{h_{out}} \;;\; T_{c_{i+1}} \equiv T_{c_{in}}$$

The above-mentioned equations are written repeatedly starting from the time when the hot stream is fed until the time that the cold stream is fed. At the boundary of the module, the specified terminal values ($T_{h_{in}}$, $T_{c_{in}}$) are used such that the degree of freedom is zero. The size of the system of ordinary differential equations (ODEs) depends on the number of divisions of the module length. Note that the total distillate production is the average of the mass production of each control volume. The above Eqs (4) and (5) are originally partial differential equations (PDEs) with advective-conduction terms in one dimension, which is the axial direction (direction of the flow). In this case, the advective-conduction terms are being upwind discretized, whereby the PDEs are reduced to a system of IVPs. The initial values of the state variables are equal to those mentioned before.

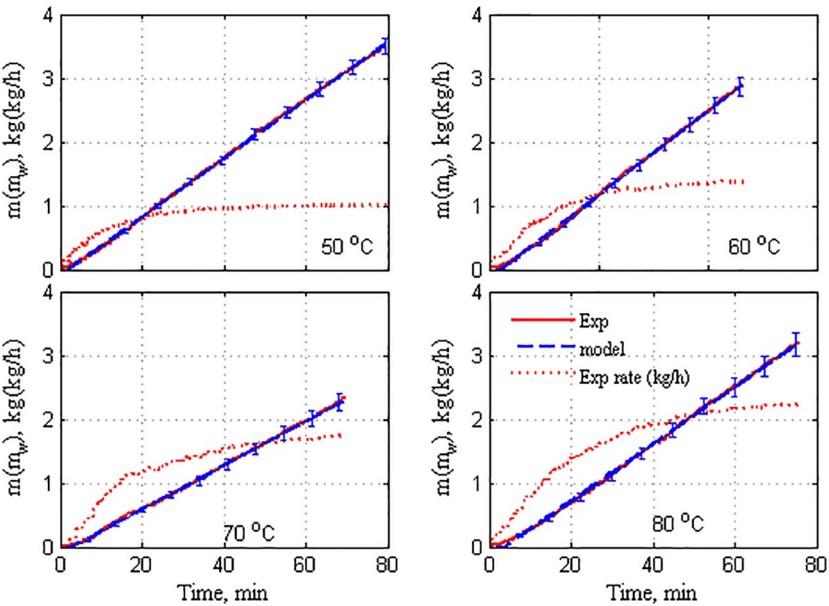

**Fig 2. Time evolution of mass production at** $m_{h_{in}} = m_{c_{in}} = 50$ **kg/h.**

## Results and discussion

The transient behavior of the accumulated mass of purified water for the selected operating conditions is depicted in Figs 2 and 3. The results for flow rates of 50 and 300 L/h are shown in the figure, but those of 100 and 200 L/h are not shown. Moreover, the trends of the responses (specified and missing) are similar albeit with different extents. The inlet cold stream temperature is maintained at 25 °C, and the inlet water salinity at 0.5% for all the experiments. The water mass is represented by a straight line because it is the measured mass of water in the

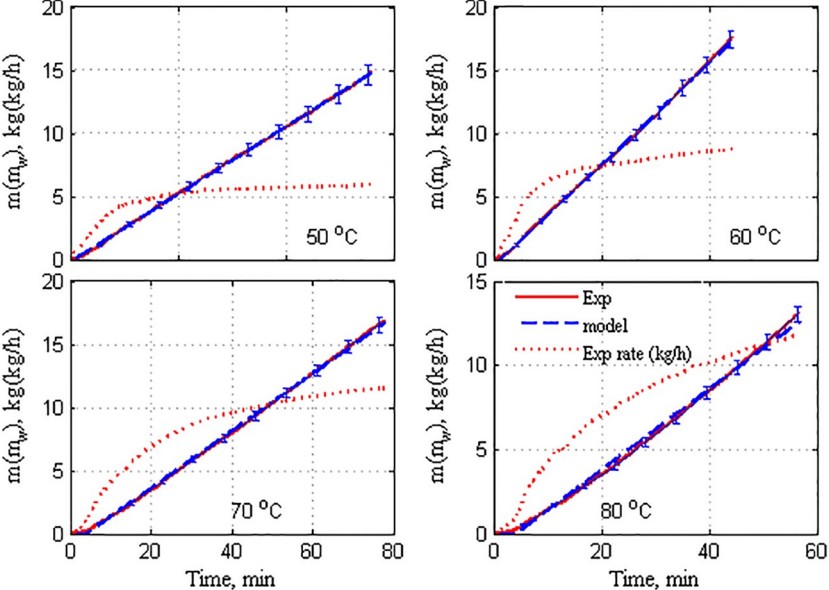

**Fig 3. Time evolution of mass production at** $m_{h_{in}} = m_{c_{in}} = 300$ **kg/h.**

collection tank at each sampling time. The error bars in the figures denote the uncertainty in the accumulated mass. The uncertainty is estimated as the standard deviation of several experimental results under identical operating conditions. Note that the error bars are placed on specific samples because their inclusion at each sampling time overfills the figure, resulting in degraded visibility. The sampling time is 10 s. The accumulated mass was transformed into the production rate and is depicted in these figures by the dotted curve. The production rate was obtained by dividing the accumulated mass by the corresponding time. The production rate is a better representation of distillate production because it clearly reveals the dynamic response. It also assists in the evaluation of the steady-state production rate. However, for certain specific operating conditions, the production rate does not attain a completely steady state. This is attributed to the fact that the experiment is terminated prematurely. The reason for the premature termination is the limited capacity of the collection tank (T3). The tank can accumulate a maximum of 25 kg of water. Beyond this, it must be emptied for sustaining continuous operation. This interruption disrupts the calculation of the mass flux over a long period. Therefore, to estimate the steady-state production rate, the latter must be extrapolated for a prolonged time. To achieve this, the measured water mass is fitted to a correlation with time. The mass accumulation at each operating condition is fitted as a straight line. The correlated mass accumulation is demonstrated using dashed lines in Figs 2 and 3. Note that the correlated collected mass is unique for each specific operating condition. That is, the collected mass for an operating condition cannot be extended for another operating condition. The aim is to obtain the perfect match of the experimental results, as evidently illustrated.

The resulting correlation is then extrapolated and transformed into the rate of change to obtain the monotone function depicted in Fig 4, for all the tested flow rates and feed temperatures. The extrapolation ensures that the mass-flux rate attains a steady state. Hence, the dynamic behavior can be analyzed conveniently, and the steady-state production rate can be estimated accurately. The process of correlation, extrapolation, and transformation into a rate of change in mass accumulation is a crucial step in this analysis. The generated asymptotic rate of mass production facilitates the dynamic analysis aimed at in this study, rather than

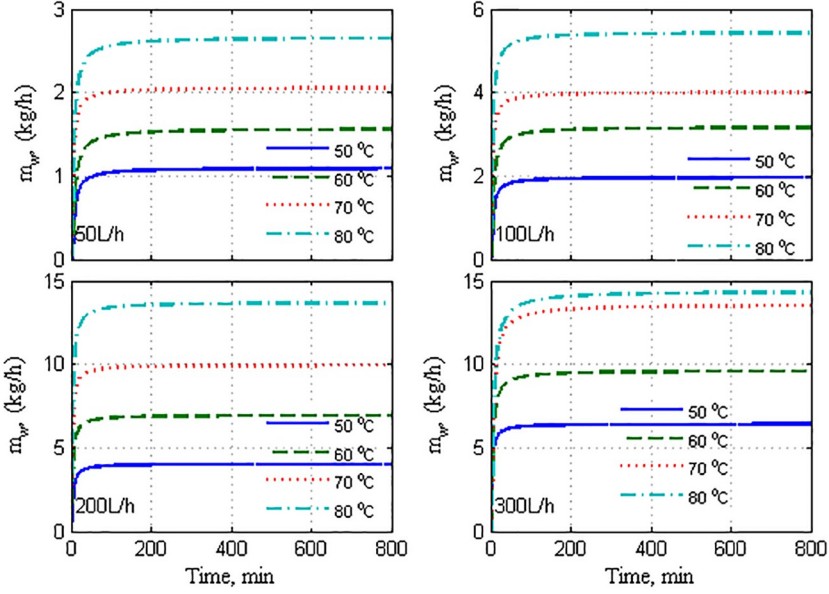

**Fig 4. Extrapolated response of mass production by using model correlation.**

integrating mass accumulation. First, the generated monotone response of the mass production can be used to determine the dynamic characteristics by using the standard reaction-curve method. Second, the generated monotone time response of the mass production can be used to validate the theoretical model given by Eqs (1)–(6). It should be noted that the dynamic model in Eqs (1–6) generates a mass production rate that exhibits an asymptotic response unlike the measured mass production, which is presented as mass accumulation.

It is apparent from Fig 4 that at each feed-flow rate, the production capacity increases proportional to the feed temperature. This observation is anticipated and reported in earlier works [7,9,41–44] because the increment in the feed temperature escalates the sensible heat of the hot stream, which results in improved heat transfer from the hot bulk side to the membrane surface. Similarly, at any operating feed temperature, the corresponding mass production rate increases with the feed-flow rate. As the feed-flow rate increases, the turbulence on the membrane surface also increases. This phenomenon improves the heat- and mass-transfer mechanisms, resulting in higher production rates [7,9,11,41–44]

It is apparent that the transient response of $m_w$ in Fig 4 resembles the reaction curve of a typical first-order system in response to stepped variations. Therefore, the linear system theory can be used to determine the dynamic characteristics, i.e., time constant and static gain, of the mass production, $m_w$. In this case, the dynamic characteristics of $m_w$ in response to the stepped variations in $T_{h_{in}}$ are estimated. The estimated time constant for the reaction curves in Fig 4 is presented in Fig 5. The measured time constant varies notably with respect to the operating conditions, indicating the nonlinearity of the process. In general, the time constant can vary between 5 and 15 min. For the lowest value of the operating feed temperature, the time constant decreases linearly as the flow rate increases. This implies that the process response becomes faster as the flow rate increases. For higher feed temperatures, the time constant decreases as the flow rate increases up to 200 L/h. Thereafter, it increases with further increases in the flow rate. The deceleration of the process dynamic at higher flow rates, specifically at 300 L/h, is not related to the fundamental mass- and heat-transfer operation inside the membrane. This behavior is ascribed to the dynamics of the external heater (H1). The reaction-curve method requires the output response to the instantaneous stepped variations in the input. However, in our experiments, the stepped variations in $T_{h_{in}}$ exhibits a certain time lag induced by the capacitance of the external heater. This underlying dynamic is marginal at small flow rates. However, it increases substantially at high flow rates and temperatures [23]. This behavior is reflected by the response speed of $m_w$ at high operating conditions.

The static mass production gain at different operating conditions is extracted from the reaction curves in Fig 4 by using the reaction-curve method. The results of the extraction are

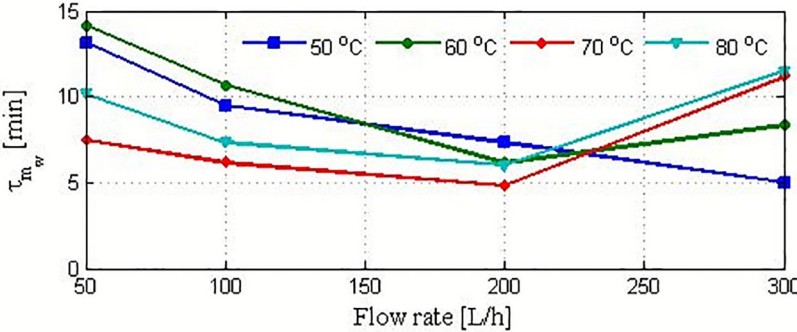

Fig 5. Time constant for mass production in response to a stepped variation in $T_{h_{in}}$.

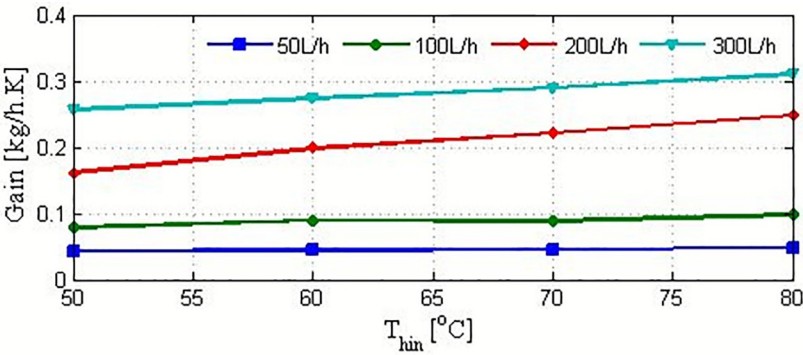

**Fig 6. Steady-state gain for $m_w$ at selected operating conditions.**

illustrated in Fig 6. These values are directly proportional to the feed-flow rate and feed temperature, as discussed previously. The static gain provides information on the effectiveness of the operating conditions. For the minimum flow rate, the static gain is almost invariant with the feed temperature. This indicates that increasing the feed temperature is ineffective for this case. Similarly, at the flow rate of 100 L/h, the static gain varies insignificantly. However, at higher flow rates, the favorable effect of the static gain on the operation is apparent. Apparently, the static gain and time constant vary considerably with operating conditions. Therefore, presentation of the mass production dynamics by a simple transfer function is not recommended. It is more effective to capture the mass production dynamics using the entire system of nonlinear ODEs, as discussed in the following section.

Further, we seek to validate the dynamic model based on the first principles expressed as Eqs (1)–(6), using the experimental data. It is to be remembered that the theoretical model differs from the straight-line correlation used previously for extrapolating the experiment data. This model is universal in that it is applicable to different operating conditions. Moreover, it is comprehensive, i.e., it explicitly includes the process parameters and other design parameters. In addition, the resultant model is a standard ODE, which is typical for dynamic and control analysis and application. The ODE model is simulated by stepping $T_{h_{in}}$ while keeping $T_{c_{in}}$ constant at 25 ˚C. The mass-flow rate is considered to be a stepwise constant, i.e., it remains constant while the IVP is solved. The IVP is solved using Euler's method with a step size of 1 s. This step size was observed to be adequate for providing a stable numerical solution. In each step of Euler's method, the intermediate parameters are determined by following the algorithm provided in Appendix B. In Fig 7, the result of the simulation is depicted and is compared to the time response of the plant. The figure depicts the simulation of the lumped model and spatial model with $n = 10$ for a wide range of operating conditions. The performance of the lumped and spatial models exhibits a notable model–plant mismatch, particularly in terms of the static gain, except at the lowest flow rate and feed temperature (50 L/h, 50 ˚C). The mismatch increases notably with flow rate and to a certain extent, with feed temperature. Whereas the lumped and spatial models delivered almost similar performance at low operating flow rates, they exhibited divergent behaviors at high flow rates. It is noteworthy that the lumped model outperformed the spatial model in terms of the overall average error. Numerically, the overall relative error for the lumped model is less than that for the spatial model, as presented in Table 1. Note that for the lumped model, the geometric average bulk temperature is used in Algorithm S1. When terminal temperatures are used as approximations of the bulk temperature in the limped model, the performance deteriorates as the overall average error attains

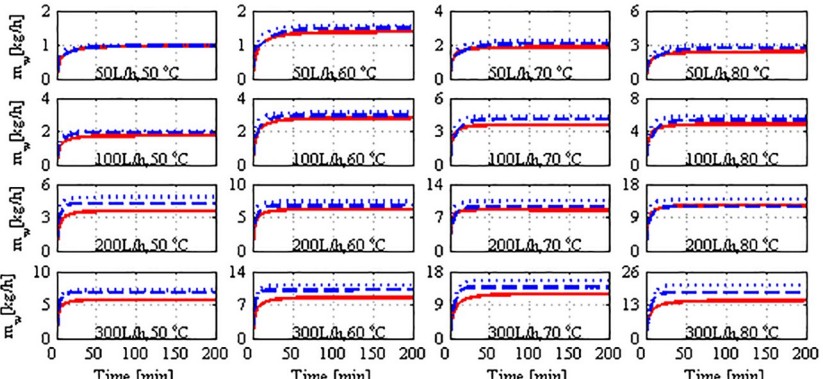

**Fig 7. Mass production response for model and plant to stepped variation; solid: Plant, dashed: Lumped model, dotted: Spatial model.**

51%, as listed in Table 1. The relative error presented in Table 1 is computed using the following formula:

$$Er = \sum_{T=50}^{80} \left( \sum_{i=1}^{n_t} \frac{m_w^m(t_i) - m_w^e(t_i)}{m_w^e(t_i)} \right)_T \times 100 \tag{7}$$

where $m_w^m$ and $m_w^e$ denote the model response and plant response, respectively, for the mass production. $n_t$ denotes the simulation time. $E_r$ is computed for each flow rate, and the overall error is the mean value across all the flow rates.

The inferiority of both the model structures can be attributed to the omission of the heat losses to the surrounding and to modeling error. Heat loss to the surroundings is reported by other works [38,44]. In general, heat losses increase with feed temperature (as manifested by the marginally increasing mismatch in Fig 7) because the difference between the module and ambient temperatures also increases. However, the increment in heat loss/modeling error with flow rate is unclear. Nevertheless, we can refer to the nonlinearity of the process observed in [32,38]. As the flow rate increases, the heat-transfer coefficients enhance the heat transfer, resulting in a higher temperature at the permeate bulk side. As the flow rate continues to increase, the heat transfer becomes asymptotic because the driving force (temperature difference between the bulk hot side and bulk cold side) decreases. Simultaneously, the temperature difference at the membrane interface increases, resulting in improved mass flux. This behavior is reflected on both the model performances when a flow rate of 200 L/h is implemented. Unlike the other cases, the model-plane discrepancy decreases with feed temperature for the

**Table 1. Percentage relative error of model–plant mismatch.**

| $m_f$ | Without tuning | | With tuning | | Without tuning and using terminal temperature |
|---|---|---|---|---|---|
| | Lumped | Spatial | Lumped | Spatial | Lumped |
| 50 L/h | 7.78 | 14.00 | 3.08 | 3.04 | 49.08 |
| 100 L/h | 10.01 | 15.97 | 3.78 | 2.90 | 46.46 |
| 200 L/h | 9.31 | 18.92 | 3.29 | 3.54 | 53.19 |
| 300 L/h | 16.99 | 25.19 | 2.71 | 3.51 | 56.51 |
| Overall | 11.02 | 18.52 | 3.22 | 3.25 | 51.31 |

**Table 2. Tuning parameter for adjusting heat loss (flos).**

|  | 50 L/h | 100 L/h | 200 L/h | 300 L/h |
|---|---|---|---|---|
| $n = 10$ | 0.15 | 0.2 | 0.22 | 0.26 |
| $n = 1$ | 0.077 | 0.11 | 0.1 | 0.17 |

case of 200 L/h. As a remedy, we propose the following model modification to incorporate the effect of heat losses to the surrounding:

For lumped case:

$$v\rho Cp\frac{dT_{h_{out}}}{dt} = m_{h_{in}}Cp\left(T_{h_{in}} - T_{rf}\right) - m_{h_{out}}Cp\left(T_{h_{out}} - T_{rf}\right) - h_mA\left(T_{h_m} - T_{c_m}\right) - j_wAH_v - f_{los}m_{h_{in}}Cp\left(T_{h_{in}} - T_{h_{out}}\right) (8)$$

For spatial case:

$$\left(\frac{v}{n}\right)\rho Cp\frac{dT_{h_i}}{dt} = m_{h_{i-1}}Cp\left(T_{h_{i-1}} - T_{rf}\right) - m_{h_i}Cp\left(T_{h_i} - T_{rf}\right) - h_{m_i}\Delta xh\left(T_{h_{m,i}} - T_{c_{m,i}}\right) - j_{w_i}\Delta xhH_v - f_{los}m_{h_{i-1}}Cp\left(T_{h_{i-1}} - T_{h_i}\right) (9)$$

Note that the heat loss term is incorporated in the thermal equation only for the hot channel. We assume that a portion of the sensible heat of the hot side is lost to the ambient. This implies that only a fraction of the hot sensible heat is available for heat transfer to the cold bulk side. The heat loss (correction term) becomes proportional to the operating temperature with the incorporation of the heat loss as a fraction of the sensible heat. This formulation mimics the modeling error observed in Fig 7, which increases with the operating temperature. To make the correction term increase with the flow rate, the fraction parameter ($f_{los}$) is set proportional to the flow rate, as presented in Table 2. The values of $f_{los}$ in Table 2 are determined by trial and error and heuristics.

Fig 8 depicts the model performance when the proposed tuning strategy is applied. It is highly evident that the model's effectiveness for both the structures is enhanced substantially. Specifically, the overall relative error is reduced to approximately 3%, as presented in Table 1. It is equally evident that the spatial model is not more advantageous than the lumped one because both structures provided remarkable fitting of the plant data when they were effectively tuned. A few marginal discrepancies are present in the time response, which are discussed in the final paragraphs. Note that the correction term in Eqs (8) and (9) is maintained

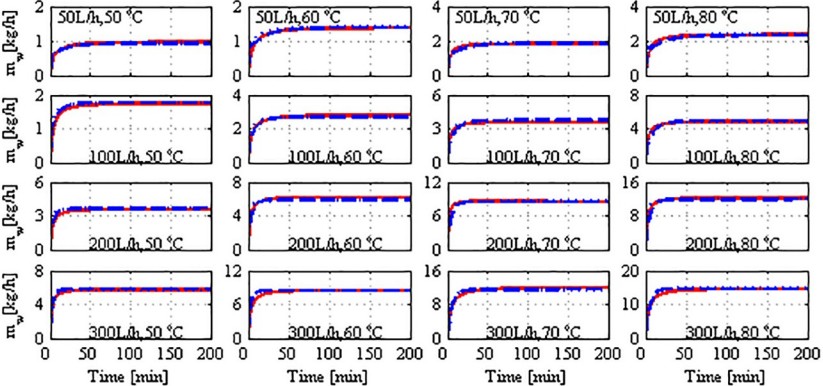

**Fig 8. Mass production response of model and plant to stepped variation with tuned $U$; solid: Plant, dashed: Lumped model, dotted: Spatial model, dash-and-dot: Independently tuned lumped model.**

as simple as possible to maintain the model simplicity and predictiveness. Although the correction term is denoted as heat losses, it appears to account for both the heat losses and other uncertain modeling errors. According to the values of $f_{los}$ in Table 2, the lumped modeling errors are 7%–17% in the lumped model and 15%–26% in the spatial model. An enhancement of the lumped model that uses terminal temperature as bulk temperature is not presented here. Our investigation revealed that tuning such a model mandates an ambiguously large correction factor ($f_{los}$).

To understand the effect of omitting the effect of heat losses on the model performance, the temperature difference at the membrane interface ($\Delta T_m = T_{h_m} - T_{c_m}$) is plotted as shown in Fig 9. This result corresponds to the case where the correction term is not incorporated in Eq (9), i.e., it belongs to the results shown in Fig 7. The interface temperature difference is relatively high compared to that shown in Fig 10, which corresponds to the case where the correction term is employed in the model. This implies that the model predicted a $\Delta T_m$ value that is larger than its true values, causing the mass production predicted by the model to inflate as shown in Fig 7. When the heat losses are omitted, the entire available sensible heat is transferred from the hot side to the cold side, whereby the permeate outlet temperature is increased. As a result, the bulk temperature difference as well as the associated interface temperature difference increase. When the heat losses term is involved, the model predicts smaller interface temperature differences, as illustrated in Fig 10. In this case, the model considers less amount of sensible heat to be transferred, resulting in a lower permeate outlet temperature and hence, smaller bulk and interface temperature differences. The smaller temperature differences generated better estimates of the mass fluxes, as depicted in Fig 8. It should be noted that the responses of $\Delta T_m$ shown in Figs 9 and 10 are for the spatial model case, where the average value of $\Delta T_m$ over the module length is plotted. A similar trend, albeit with a different extent, is obtained for the lumped model case. The results are not shown here because Figs 9 and 10 serve the purpose. Regardless of the model structure used, the trend of $\Delta T_m$ response coincides

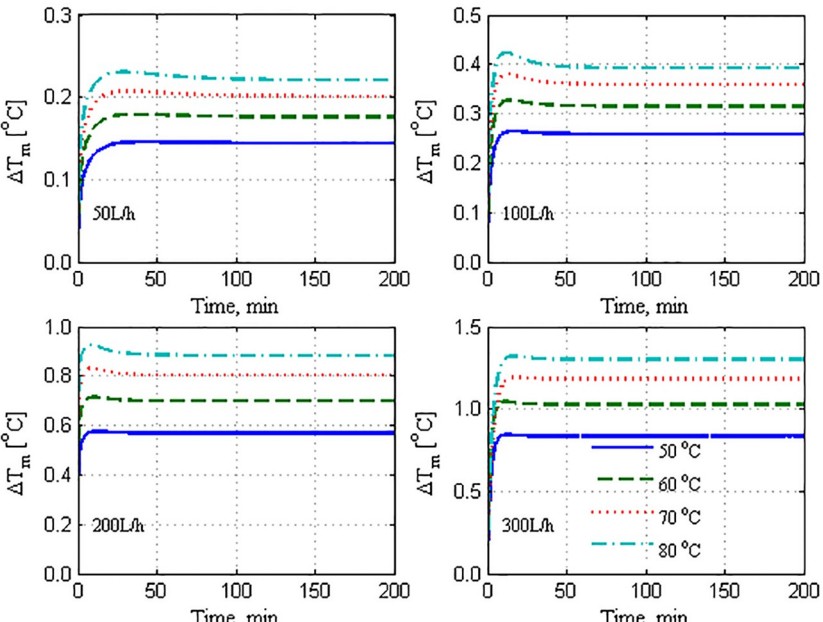

**Fig 9. Average difference between hot and cold membrane interface temperatures omitting heat losses.**

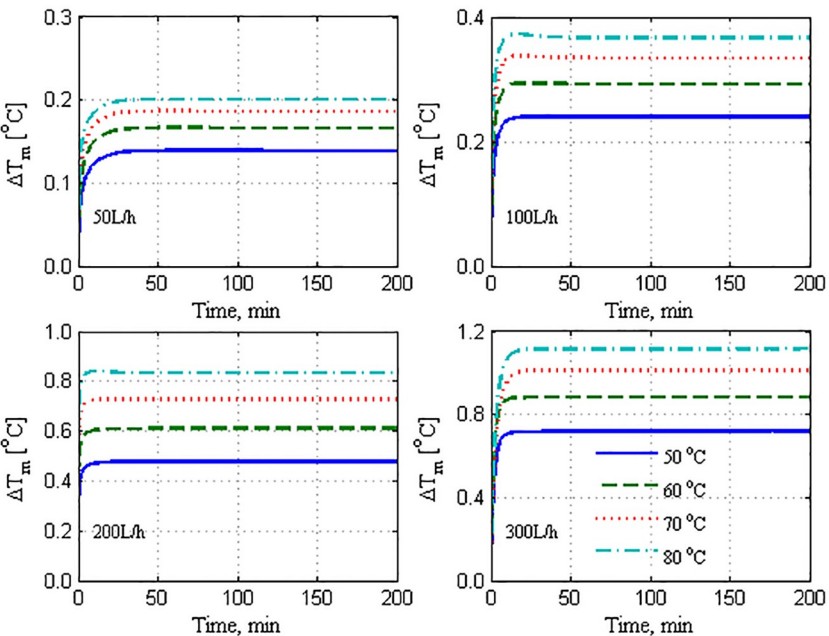

**Fig 10. Average difference between hot and cold membrane interface temperatures considering heat losses.**

with that of the mass production shown in Fig 4, wherein it is proportional to the hot feed temperature and operating flow rate.

The proposition of heat losses to the surroundings is reasonable and has been observed by others as mentioned earlier. However, the proportionality of the heat losses to the flow rate is unclear. Another method to improve the model performance through a better estimation of the heat transfer mechanism is also valid. Figs 7 and 9 show that the values of the mass flux predicted by the model are higher than the measured ones because of the erroneous estimation of the interface temperatures. The latter are estimated by Eqs (B.12) and (B.13). Hence, the erroneous values of the interface temperatures could be owing to the limitation of the model in adapting the values of $h_p$ and $h_f$ or $U$ to variations in the operating conditions. Therefore, a better estimation of the interface temperatures and consequently the mass flux can be achieved by penalizing $U$ directly or using better Nusselt correlation. Investigation on this by the authors is underway.

It should be reminded that Algorithm S1 is still applied in the numerical solution of the IVP (Eqs (1)–(6)) because the algorithm estimates internal parameters of the model, such as the membrane coefficient, $C_m$, membrane-surface temperatures, $T_{h_m}$, $T_{c_m}$, and local heat-transfer coefficients, $h_f$, $h_p$. These parameters are embedded in the IVP problem. Nevertheless, the origin of the model–plant mismatch exhibited in the model performance should be highlighted here. This versatile mismatch is the result of several factors such as uncertainty in process measurements, uncertainty owing to the modeling simplifications, limitation of the process design, and nonlinearity of the process. The model should not be constrained to match the experimental data because the measured data contain certain uncertainties. The plant experiments exhibited both measurement and random errors, as illustrated in Figs 2 and 3. The models were simulated using identical initial values, whereas the actual initial temperatures in the experiments varied across the tests by ±1–3˚ depending upon the room temperature. The stepped variations in $T_{h_{in}}$ in the real plant are not instantaneous and rather exhibit a dynamic behavior. $T_{c_{in}}$ is not constant during the experiments. It exhibits randomness owing

to the manual-control system. This issue is particularly discussed in a previously published work [32].

## Conclusions

In this study, a first-principles dynamic model for purified water production using the DCMD process is developed. Specifically, the structures of both the lumped-parameter model and spatial model were constructed. The accuracy of both the models in predicting the distillate rate was tested against experimental data generated by step testing a DCMD pilot plant. The accuracy of both the model structures was low, with the lumped-model structure being marginally superior to the spatial model. Although the underlying parameters of both the models were calculated at each sampling time by solving combined mass and energy balances, the models could not effectively track the mass production response over a wide range of operating conditions. An adjustment of the model by incorporating a heat loss term improved their performances: the overall relative error reduced to approximately 3%. Alternatively, the heat transfer correlation could be revised for enhanced model performance.

## Appendix A

Considering the whole MD unit as a lumped-parameter system as shown in S1 Fig, the steady state mass production rate of water is expressed as follows [32]:

$$m_w = C_m(P_1 - P_2)A \tag{A.1}$$

$P_1$ and $P_2$ are the partial pressures of water vapor estimated at the membrane surface temperatures $T_{hm}$ and $T_{cm}$, respectively. These can be calculated using the Antoine equation [6,7]:

$$P_1 = \exp(a - \frac{b}{T_{h_m} - c}) \tag{A.2}$$

$$P_2 = \exp(a - \frac{b}{T_{c_m} - c}) \tag{A.3}$$

The membrane interface temperatures can be related to the membrane bulk temperature when the convection and conduction heat transfer attain equilibrium [8]:

$$h_m(T_{h_m} - T_{c_m}) + j_w H_v = h_f(T_{h_b} - T_{h_m}) = U(T_{h_b} - T_{c_b}) \tag{A.4}$$

$$h_m(T_{h_m} - T_{c_m}) + j_w H_v = h_p(T_{c_m} - T_{c_b}) = U(T_{h_b} - T_{c_b}) \tag{A.5}$$

Note that the equalities (A.4) and (A.5) hold at steady state. Because we are developing a dynamic model, we assume that they hold at a pseudo-steady state. Therefore, from the left side of the last two equations, i.e., by equating the membrane conduction plus latent heat terms to the convection from the bulk to the membrane surface, we obtain

$$T_{h_m} = \frac{h_f(h_m + h_p)}{h_p h_f + h_m(h_p + h_f)} T_{h_b} + \frac{h_m h_p}{h_p h_f + h_m(h_p + h_f)} T_{c_b} - (\frac{h_p}{h_p h_f + h_m(h_p + h_f)}) j_w H_v \tag{A.6}$$

$$T_{c_m} = \frac{h_f h_m}{h_p h_f + h_m(h_p + h_f)} T_{h_b} + \frac{h_p(h_m + h_f)}{h_p h_f + h_m(h_p + h_f)} T_{c_b} + \frac{h_f}{h_p h_f + h_m(h_p + h_f)} j_w H_v \tag{A.7}$$

or in a compact form as

$$T_{h_m} = a_h \ T_{h_b} + a_c \ T_{c_b} - c_h \ j_w H_v \qquad \text{(A.6a)}$$

$$T_{c_m} = b_h T_{h_b} + b_c T_{c_b} + c_c \ j_w H_v \qquad \text{(A.7a)}$$

where

$$a_h = \frac{h_f(h_m + h_p)}{h_p h_f + h_m(h_p + h_f)} \qquad \text{(A.8)}$$

$$a_c = \frac{h_p h_m}{h_p h_f + h_m(h_p + h_f)} \qquad \text{(A.9)}$$

$$b_h = \frac{h_f h_m}{h_p h_f + h_m(h_p + h_f)} \qquad \text{(A.10)}$$

$$b_c = \frac{h_p(h_m + h_f)}{h_p h_f + h_m(h_p + h_f)} \qquad \text{(A.11)}$$

$$c_h = \left[ \frac{h_p}{h_p h_f + h_m(h_p + h_f)} \right] \qquad \text{(A.12)}$$

$$c_c = \left[ \frac{h_f}{h_p h_f + h_m(h_p + h_f)} \right] \qquad \text{(A.13)}$$

$C_m$ is the MD coefficient and is considered a weak function of temperature while deriving the dynamic model. Further details on computing $C_m$ are provided in Appendix B. Similarly, the physical properties of water are a function of temperature. However, we omit their derivative with respect to temperature. Hence, the derivative of (A.1) with respect to time is

$$\frac{dm_w}{dt} = C_m A \frac{d}{dt}(P_1 - P_2). \qquad \text{(A.14)}$$

The above equation was presented by Hassan et al. [34]. However, they did not consider calculating the derivative of the vapor pressures. In the following, we present the derivation of the partial derivatives to obtain a complete expression for the time evolution of the transmembrane flux:

$$\frac{dm_w}{dt} = C_m A \left( \frac{\partial P_1}{\partial T_{h_m}} \frac{\partial T_{h_m}}{\partial T_{h_b}} \frac{dT_{h_b}}{dt} + \frac{\partial P_1}{\partial T_{h_m}} \frac{\partial T_{h_m}}{\partial T_{c_b}} \frac{dT_{c_b}}{dt} + \frac{\partial P_1}{\partial T_{h_m}} \frac{\partial T_{h_m}}{\partial m_w} \frac{dm_w}{dt} - \frac{\partial P_2}{\partial T_{c_m}} \frac{\partial T_{c_m}}{\partial T_{h_b}} \frac{dT_{h_b}}{dt} - \frac{\partial P_2}{\partial T_{P_m}} \frac{\partial T_{c_m}}{\partial T_{c_b}} \frac{dT_{c_b}}{dt} \frac{\partial P_2}{\partial T_{c_m}} \frac{\partial T_{c_m}}{\partial m_w} \frac{dm_w}{dt} \right) \text{(A.14a)}$$

The partial derivatives in Eq (A.14) can be obtained as follows. The derivatives of the partial pressure given in Eqs (A.2) and (A.3), with respect to the membrane interface temperatures

are

$$\frac{\partial P_1}{\partial T_{h_m}} = \frac{b}{\left(T_{h_m} - c\right)^2} \exp\left(a - \frac{b}{T_{h_m} - c}\right) \equiv \beta_h \tag{A.15}$$

$$\frac{\partial P_2}{\partial T_{c_m}} = \frac{b}{\left(T_{c_m} - c\right)^2} \exp\left(a - \frac{b}{T_{c_m} - c}\right) \equiv \beta_c \tag{A.16}$$

The derivatives of the membrane interface temperature given in Eqs (A.6a) and (A.7a), with respect to the bulk temperature and mass flux are

$$\frac{\partial T_{h_m}}{\partial T_{h_b}} = a_h; \quad \frac{\partial T_{h_m}}{\partial T_{c_b}} = a_c \ ; \quad \frac{\partial T_{h_m}}{\partial m_w} = -c_h H_v / A \tag{A.17}$$

$$\frac{\partial T_{c_m}}{\partial T_{h_b}} = b_h; \quad \frac{\partial T_{c_m}}{\partial T_{c_b}} = b_c \ ; \quad \frac{\partial T_{c_m}}{\partial m_w} = c_c H_v / A \tag{A.18}$$

Substituting Eqs (A.15)–(A.18) into Eq (A.14) yields

$$\frac{dm_w}{dt} = \frac{C_m A}{\left(1 + C_m \beta_h c_h H_v + C_m \beta_c c_c H_v\right)} \left(\left(\beta_h a_h - \beta_c b_h\right) \frac{dT_{h_b}}{dt} + \left(\beta_h a_c - \beta_c b_c\right) \frac{dT_{c_b}}{dt}\right) \tag{A.19}$$

The dynamic model for the mass production in Eq (A.19) depends explicitly on the dynamics of the bulk temperature in the hot and cold sides. The dynamic of the thermal behavior of the MD was developed by [32] and is as follows:

$$\rho Cp \frac{dT_{h_{out}}}{dt} = m_{h_{in}} Cp\left(T_{h_{in}} - T_{rf}\right) - m_{h_{out}} Cp\left(T_{h_{out}} - T_{rf}\right) - h_m A\left(T_{h_m} - T_{c_m}\right) - j_w A H_v \tag{A.20}$$

$$v\rho Cp \frac{dT_{c_{out}}}{dt} = m_{c_{in}} Cp\left(T_{c_{in}} - T_{rf}\right) - m_{c_{out}} Cp\left(T_{c_{out}} - T_{rf}\right) + h_m A\left(T_{h_m} - T_{c_m}\right) + j_w A H_v \tag{A.21}$$

The mass balance equations are as follows:

$$m_{h_{out}} = m_{h_{in}} - m_w \tag{A.22}$$

$$m_{c_{out}} = m_{c_{in}} + m_w \tag{A.23}$$

For the lumped-parameter model, Eq (A.19) requires the time-derivative of the bulk temperature, whereas the thermal Eqs (A.20) and (A.21) are defined in terms of the terminal temperature. Hence, to completely specify the IVPs, i.e., link the three state variables, we assume the following relationship between the bulk and boundary temperatures:

$$T_{h_b} = \sqrt{T_{h_{in}} T_{h_{out}}} \ ; \quad T_{c_b} = \sqrt{T_{c_{in}} T_{c_{out}}} \tag{A.24}$$

The derivatives of the last equations with respect to time are

$$\frac{dT_{h_b}}{dt} = \frac{1}{2} \sqrt{\frac{T_{h_{in}}}{T_{h_{out}}}} \frac{dT_{h_{out}}}{dt} \tag{A.25}$$

$$\frac{dT_{c_b}}{dt} = \frac{1}{2} \sqrt{\frac{T_{c_{in}}}{T_{c_{out}}}} \frac{dT_{c_{out}}}{dt} \tag{A.26}$$

Inserting Eqs (A.25) and (A.26) into Eq (A.19) yields

$$\frac{dm_w}{dt} = \frac{C_m A}{(1 + C_m \beta_h c_h H_v + C_m \beta_c c_c H_v)} \left( (\beta_h a_h - \beta_c b_h) \frac{1}{2} \sqrt{\frac{T_{h_{in}}}{T_{h_{out}}}} \frac{dT_{h_{out}}}{dt} + (\beta_h a_c - \beta_c b_c) \frac{1}{2} \sqrt{\frac{T_{c_{in}}}{T_{c_{out}}}} \frac{dT_{c_{out}}}{dt} \right) \tag{A.27}$$

Hence, Eqs (A.20)–(A.23) and (A.27) define the complete dynamic model for the mass production in the lumped form. In this study, the inlet flow rates for the cold and hot sides are equal, i.e., $m_{c_{in}} = m_{h_{in}}$. The operation is limited by the experimental procedure. Note that the membrane surface temperatures, $T_{h_m}$, $T_{c_m}$, internal heat transfer coefficients, $h_f$ $h_p$, $h_m$, and intermediate variables, $j_w$, $H_v$ are obtained by solving the combined mass and heat transfer equations as described in Appendix B. These parameters are determined at each time-instant during the numerical solution of the IVP (Eqs A.20, A.21 and A.27).

## Extension of lumped dynamic model to one-dimensional formulation

To develop the axial dynamic model, the thermal balance represented by Eqs (A.20) and (A.21) can be written for a specific control volume of the membrane, as shown in S1 Fig. Hence, by discretizing Eqs (A.20) and (A.21), the resulting system of IVPs is expressed as follows:

$$\left(\frac{v}{n}\right)\rho Cp \frac{dT_{h_i}}{dt} = m_{h_{i-1}} Cp\left(T_{h_{i-1}} - T_{rf}\right) - m_{h_i} Cp\left(T_{h_i} - T_{rf}\right) - h_{m_i} \Delta xh\left(T_{h_{m,i}} - T_{c_{m,i}}\right) - j_{w_i} \Delta xh H_v \tag{A.28}$$

$$\left(\frac{v}{n}\right)\rho Cp \frac{dT_{c_i}}{dt} = m_{c_{i+1}} Cp\left(T_{c_{i+1}} - T_{rf}\right) - m_{c_i} Cp\left(T_{c_i} - T_{rf}\right) + h_{m_i} \Delta xh\left(T_{h_{m,i}} - T_{c_{m,i}}\right) + j_{w_i} \Delta xh H_v \tag{A.29}$$

$$i = 1, \cdots, n$$

$$for\ i = 1\ \rightarrow T_{h_{i-1}} \equiv T_{h_{in}}\ ;\ T_{c_i} \equiv T_{c_{out}}$$

$$for\ i = n\ \rightarrow T_{h_i} \equiv T_{h_{out}}\ ;\ T_{c_{i+1}} \equiv T_{c_{in}}$$

The corresponding infinitesimal dynamic model for the mass production (A.19) can be expressed as follows:

$$\frac{dm_{w_i}}{dt} = \frac{C_{m_i} A}{(1 + C_{m_i} \beta_{h_i} c_{h_i} H_{v_i} + C_{m_i} \beta_{c_i} c_{c_i} H_{v_i})} \left( \left(\beta_{h_i} a_{h_i} - \beta_{c_i} b_{h_i}\right) \frac{dT_{h_i}}{dt} + \left(\beta_{h_i} a_{c_i} - \beta_{c_i} b_{c_i}\right) \frac{dT_{c_i}}{dt} \right) \tag{A.30}$$

where

$$\beta_{h_i} = \frac{b}{(T_{hm,i} - c)^2} \exp\left(a - \frac{b}{T_{hm,i} - c}\right) \tag{A.31}$$

$$\beta_{c_i} = \frac{b}{(T_{cm,i} - c)^2} \exp\left(a - \frac{b}{T_{cm,i} - c}\right) \tag{A.32}$$

$$a_{h_i} = \frac{h_{f_i}(h_{m_i} + h_{p_i})}{h_{p_i} h_{f_i} + h_{m_i}(h_{p_i} + h_{f_i})} \tag{A.33}$$

$$a_{c_i} = \frac{h_{p_i} h_{m_i}}{h_{p_i} h_{f_i} + h_{m_i}(h_{p_i} + h_{f_i})} \tag{A.34}$$

$$b_{h_i} = \frac{h_{f_i} h_{m_i}}{h_{p_i} h_{f_i} + h_{m_i}(h_{p_i} + h_{f_i})} \tag{A.35}$$

$$b_{c_i} = \frac{h_{p_i}(h_{m_i} + h_{f_i})}{h_{p_i} h_{f_i} + h_{m_i}(h_{p_i} + h_{f_i})} \tag{A.36}$$

$$c_{h_i} = \left[\frac{h_{p_i}}{h_{p_i} h_{f_i} + h_{m_i}(h_{p_i} + h_{f_i})}\right] \tag{A.37}$$

$$c_{c_i} = \left[\frac{h_{f_i}}{h_{p_i} h_{f_i} + h_{m_i}(h_{p_i} + h_{f_i})}\right] \tag{A.38}$$

whereas $h_{f_i}$, $h_{p_i}$, $h_{m_i}$, $C_{m_i}$ are calculated from (B.1, B.11, B.4–B.6) using the bulk temperature for each i$^{\text{th}}$ control volume.

## Appendix B

The overall heat-transfer coefficient, membrane interface temperatures, and membrane coefficient are calculated by the following algorithm (S1). The following formulation is the conclusion of our previously conducted work on MD modeling and analysis [32,45–47].

1. Given the bulk temperatures at both sides of the MD membrane, $T_{h_b}$, $T_{c_b}$, the local heat-transfer coefficients, $h_f, h_p$, are calculated from the Nusselt number as follows (Alkhudairi et al., 2012):

$$Nu = 0.298 Re^{n_1} Pr^{n_2} \tag{B.1}$$

where Re denotes the Reynolds number, Pr denotes the Prandtl number, $n_1 = 0.646$, and $n_2 = 0.316$.
For the lumped-parameter model, the inlet temperature of the hot stream ($T_{h_{in}}$) is generally approximated as the bulk temperature of the hot stream, whereas the outlet temperature of the cold stream ($T_{c_{out}}$) is approximated as that of the cold stream. The use of boundary

temperatures for the bulk results in higher values of $m_w$ in the steady state [32]. Alternatively, the geometric mean of the boundary temperatures (A.25) and (A.26) can be used. For the spatial model, the bulk temperatures are approximated as the bulk temperature of each control volume.

2. Set $T_h^0 m = T_{h_b}$ and $T_c^0 m = T_{c_b}$

3. Calculate the vapor pressure at the membrane interface using the following [6]:

$$P_1 = \exp(23.238 - \frac{3841}{T_{hm} - 45})(1 - C_s)(1 - 0.5C_s - 10C_s^2) \tag{B.2}$$

$$P_2 = \exp(23.238 - \frac{3841}{T_{cm} - 45}) \tag{B.3}$$

4. After determining the membrane characteristics and average membrane temperature, i.e., $T = \frac{T_{hm} + T_{cm}}{2}$, the membrane coefficient $C_m$ can be estimated using the correlation provided in [8] according to the designated mechanism as follows:

- Knudson-flow mechanism, $k_n > 1$:

$$C_m^k = \frac{2\varepsilon r}{3\tau\delta}\left(\frac{8M_w}{\pi RT}\right)^{1/2} \tag{B.4}$$

- Molecular-diffusion mechanism, $k_n < 0.01$:

$$C_m^D = \frac{\varepsilon}{\tau\delta}\frac{PD}{P_a}\frac{M_w}{RT} \tag{B.5}$$

- Knudsen molecular-diffusion transition mechanism, $0.01 < k_n < 1$:

$$C_m^C = \left[\frac{3}{2}\frac{\tau\delta}{\varepsilon r}\left(\frac{\pi RT}{8M_w}\right)^{1/2} + \frac{\tau\delta}{\varepsilon}\frac{P_a}{PD}\frac{RT}{M_w}\right]^{-1} \tag{B.6}$$

where the Knudsen number is defined as $k_n = \frac{\lambda}{d}$. $\lambda$ is the mean free path of water molecules and is expressed as [4]:

$$\lambda = \frac{k_B T}{\sqrt{2}\pi P d_e^2} \tag{B.7}$$

5. Calculate the latent heat of vaporization at the average membrane temperature as

$$H_v(T) = 1850.7 + 2.8273T - 1.6 \times 10^{-3} T^2 \tag{B.8}$$

6. Calculate the mass flux as

$$j_w = C_m(P_1 - P_2) \tag{B.9}$$

7. Compute the overall heat-transfer coefficient using [8]

$$U = \left[\frac{1}{h_f} + \frac{1}{h_m + \frac{JH_v}{T_{h_m} - T_{c_m}}} + \frac{1}{h_p}\right]^{-1} \tag{B.10}$$

The membrane heat-transfer coefficient ($h_m$) represents the heat resistance owing to conduction and can be estimated using [11]

$$h_m = \frac{k_m}{\delta} = \frac{(1 - \varepsilon)k_s + \varepsilon k_g}{\delta} \tag{B.11}$$

8. At equilibrium, all the heat-transfer mechanisms within the MD are equivalent. Hence, Eqs (A.4) and (A.5) hold. The membrane interface temperatures are computed using the right-side equality of Eqs (A.6) and (A.7):

$$h_f(T_{h_b} - T_{h_m}) = U(T_{h_b} - T_{c_b}) \tag{B.12}$$

$$h_p(T_{c_m} - T_{c_b}) = U(T_{h_b} - T_{c_b}) \tag{B.13}$$

9. If $T_{h_m} = T^0_{h_m}$ and $T_{c_m} = T^0_{c_m}$, stop the iteration; otherwise, set $T^0_{h_m} = T_{h_m}$ and $T^0_{c_m} = T_{c_m}$. Then, return to Step 3.

The above-mentioned algorithm is terminated using a termination tolerance of $1 \times 10^{-7}$. Note that the algorithm is applicable to both the lumped and spatial versions of the model. Furthermore, for both models, the above-mentioned algorithm is solved assuming quasi-steady-state conditions.

## Nomenclature

| | |
|---|---|
| $A$ | Cross-sectional area, m$^2$ |
| $A$ | Antione equation constant |
| $b$ | Antione equation constant |
| $c$ | Antione equation constant |
| $C_m$ | Permeability coefficient, kg/m$^2$·s·Pa |
| $C_m^k$ | Knudsen mass-flux coefficient, kg/m$^2$·s·Pa |
| $C_m^d$ | Molecular diffusion mass-flux coefficient, kg/m$^2$·s·Pa |
| $C_m^C$ | Transition mass-flux coefficient, kg/m$^2$·s·Pa |
| $C_p$ | Heat capacity, J/kg·K |
| $C_s$ | Salt concentration, % |
| $E_r$ | Percentage relative error for model–plant mismatch |
| $H_v$ | Latent heat of vaporization, J/kg |

*(Continued)*

(Continued)

| | |
|---|---|
| $h_f, h_p, h_m$ | Heat-transfer coefficients of the feed, permeate, and membrane, W/m$^2$·K |
| $H$ | Channel height, m |
| $J_w$ | Mass flux, kg/m$^2$·h |
| $k_1, k_2$ | Static gains |
| $k_B$ | Boltzmann constant |
| $k_m$ | Membrane conductivity, W/m·K |
| $k_s$ | Solid-phase thermal conductivity, W/m·K |
| $k_g$ | Gas-phase thermal conductivity, W/m·K |
| $k_n$ | Knudsen number |
| $m_{h_{in}}, m_{h_{out}}$ | Inlet and outlet mass-feed rate for hot fluid, kg/h |
| $m_{c_{in}}, m_{c_{out}}$ | Inlet and outlet mass-feed rate for cold fluid, kg/h |
| $m_{h_i}$ | Mass-flow rate for hot side for i$^{th}$ cell, kg/h |
| $m_{c_i}$ | Mass-flow rate for cold side for i$^{th}$ cell, kg/h |
| $m$ | Distillate mass, kg |
| $m_w$ | Distillate flow rate, kg/h |
| $M_w$ | Molecular weight |
| $Nu$ | Nusselt number |
| $N$ | Number of membrane-length divisions, i.e., control elements |
| $n_t$ | Length of simulation time |
| $P_1, P_2$ | Vapor pressure at feed and permeate membrane surface, Pa |
| $P_a$ | Entrapped-air pressure, Pa |
| PD | Membrane pressure multiplied by diffusivity, Pa·m$^2$/s |
| Pr | Prandtl number |
| $R$ | Membrane pore size, m |
| R | Ideal gas constant, |
| Re | Reynold number |
| S | Laplace domain |
| $T$ | Time |
| $T_h, T_c$ | Feed (hot) and permeate (cold) temperatures, K |
| $T_{hb}, T_{cb}$ | Feed (hot) and permeate (cold) bulk temperatures, K |
| $T_{hm}, T_{cm}$ | Feed and permeate membrane temperatures, K |
| $T_{h_{out}}, T_{h_{in}}$ | Outlet and inlet hot feed temperatures, ˚C |
| $T_{c_{out}}, T_{c_{in}}$ | Outlet and inlet cold stream temperatures, ˚C |
| TF | Transfer function |
| $U$ | Overall heat-transfer coefficient, W/m$^2$·K |
| $V$ | Channel volume, m$^3$ |
| *Greek letters* | |
| T | Time constant, min, also membrane tortuosity |
| ρ | Water density, kg/m$^3$ |
| δ | Membrane thickness |
| ε | Porosity |
| Λ | Mean free path, m |

## Supporting information

**S1 Fig. Schematic of MD process: (a) lumped module, (b) i$^{th}$ control volume of MD module.**
(TIF)

**S1 Data.**
(XLSX)

**S2 Data.**
(XLSX)

**S3 Data.**
(XLSX)

**S4 Data.**
(XLSX)

**S5 Data.**
(XLSX)

## Author Contributions

**Conceptualization:** Emad Ali.

**Data curation:** Abdullah Najib.

**Formal analysis:** Emad Ali, Jamel Orfi.

**Investigation:** Jamel Orfi, Abdullah Najib.

**Methodology:** Emad Ali.

**Software:** Emad Ali.

**Supervision:** Jamel Orfi.

**Validation:** Abdullah Najib.

**Visualization:** Abdullah Najib.

**Writing – original draft:** Emad Ali, Jamel Orfi.

**Writing – review & editing:** Emad Ali, Jamel Orfi.

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
