## [Decision Letter · Decision Letter 0]

24 Dec 2019

PONE-D-19-31057

Developing and Validating a Dynamic Model of Water Production in Direct-Contact Membrane Distillation

PLOS ONE

Dear Prof Ali,

Thank you for submitting your manuscript to PLOS ONE. After careful consideration, we feel that it has merit but does not fully meet PLOS ONE’s publication criteria as it currently stands. Therefore, we invite you to submit a revised version of the manuscript that addresses the points raised during the review process.

We would appreciate receiving your revised manuscript by Feb 07 2020 11:59PM. To enhance the reproducibility of your results, we recommend that if applicable you deposit your laboratory protocols in protocols.io, where a protocol can be assigned its own identifier (DOI) such that it can be cited independently in the future. For instructions see: http://journals.plos.org/plosone/s/submission-guidelines#loc-laboratory-protocols

We look forward to receiving your revised manuscript.

Kind regards,

Liao Zuwei

Academic Editor

PLOS ONE

Journal Requirements:

Reviewers' comments:

Reviewer's Responses to Questions

**Comments to the Author**

1. Is the manuscript technically sound, and do the data support the conclusions?

Reviewer #1: Partly

Reviewer #2: Yes

2. Has the statistical analysis been performed appropriately and rigorously? 

Reviewer #1: No

Reviewer #2: Yes

3. Have the authors made all data underlying the findings in their manuscript fully available?

Reviewer #1: Yes

Reviewer #2: Yes

4. Is the manuscript presented in an intelligible fashion and written in standard English?

Reviewer #1: No

Reviewer #2: No

5. Review Comments to the Author

Reviewer #1: 1. The Experimental devices were same as the reference [32,35,38,39], what’s the main difference between this paper and the reference [32,35,38,39]. The innovations should be clarified.

2. Adjustment of the model by adding heat loss term resulted in enhanced predictions, but the adjustment is also based on regression and fitting. So the universalities of the modified model need to be discussed. And there were no theoretical explanations. The Experimental devices were a MD pilot plant, and the scale-up problems are unavoidable when the MD is applied in practical industries, then the adjustment may not be applicable. And carrying out experiments in practical industries is not realistic.

3. The Figures are not clear enough to be published. Legends were missed in some Figures.

Reviewer #2: 1. The experimental results and analysis did not clearly indicate the advantages and significance of the spatial model constructed in this manuscript.

2. It is recommended to mark the legends in each Figure to make the physical meaning of each line more intuitive.

3. In general, there is a lack of explanation of the lumped model slightly better than the spatial model.

4. The overall idea of the article is clear, but the appendix is relatively long. It is recommended to include part of the appendix in the main text.

5. It is noted that your manuscript needs careful editing by someone with expertise in technical English editing paying particular attention to English grammar, spelling and sentence structure so that the goals and result of the study are clear to the reader.

6. PLOS authors have the option to publish the peer review history of their article (what does this mean?). If published, this will include your full peer review and any attached files.

Reviewer #1: No

Reviewer #2: No

---

## [Decision Letter · Decision Letter 1]

25 Feb 2020

Developing and Validating a Dynamic Model of Water Production in Direct-Contact Membrane Distillation

PONE-D-19-31057R1

Dear Dr. Ali,

We are pleased to inform you that your manuscript has been judged scientifically suitable for publication and will be formally accepted for publication once it complies with all outstanding technical requirements.

With kind regards,

Zuwei Liao

Academic Editor

PLOS ONE

Additional Editor Comments (optional):

Reviewers' comments:

Reviewer's Responses to Questions

**Comments to the Author**

1. If the authors have adequately addressed your comments raised in a previous round of review and you feel that this manuscript is now acceptable for publication, you may indicate that here to bypass the “Comments to the Author” section, enter your conflict of interest statement in the “Confidential to Editor” section, and submit your "Accept" recommendation.

Reviewer #1: All comments have been addressed

Reviewer #2: All comments have been addressed

2. Is the manuscript technically sound, and do the data support the conclusions?

Reviewer #1: Yes

Reviewer #2: (No Response)

3. Has the statistical analysis been performed appropriately and rigorously? 

Reviewer #1: Yes

Reviewer #2: (No Response)

4. Have the authors made all data underlying the findings in their manuscript fully available?

Reviewer #1: Yes

Reviewer #2: (No Response)

5. Is the manuscript presented in an intelligible fashion and written in standard English?

Reviewer #1: Yes

Reviewer #2: (No Response)

6. Review Comments to the Author

Reviewer #1: (No Response)

Reviewer #2: (No Response)

7. PLOS authors have the option to publish the peer review history of their article (what does this mean?). If published, this will include your full peer review and any attached files.

Reviewer #1: No

Reviewer #2: No

---

## [Editor Report · Acceptance letter]

9 Mar 2020

PONE-D-19-31057R1 

Developing and Validating a Dynamic Model of Water Production by Direct-Contact Membrane Distillation 

Dear Dr. Ali:

I am pleased to inform you that your manuscript has been deemed suitable for publication in PLOS ONE. Congratulations! Your manuscript is now with our production department. 

With kind regards,

on behalf of

Dr. Zuwei Liao 

Academic Editor

PLOS ONE